# A Mini-Review on Reflectins, from Biochemical Properties to Bio-Inspired Applications

**DOI:** 10.3390/ijms232415679

**Published:** 2022-12-10

**Authors:** Junyi Song, Baoshan Li, Ling Zeng, Zonghuang Ye, Wenjian Wu, Biru Hu

**Affiliations:** College of Science, National University of Defense Technology, Changsha 410073, China

**Keywords:** reflectin, origin, biochemical properties, self-organization, bio-inspired applications

## Abstract

Some cephalopods (squids, octopuses, and cuttlefishes) produce dynamic structural colors, for camouflage or communication. The key to this remarkable capability is one group of specialized cells called iridocytes, which contain aligned membrane-enclosed platelets of high-reflective reflectins and work as intracellular Bragg reflectors. These reflectins have unusual amino acid compositions and sequential properties, which endows them with functional characteristics: an extremely high reflective index among natural proteins and the ability to answer various environmental stimuli. Based on their unique material composition and responsive self-organization properties, the material community has developed an impressive array of reflectin- or iridocyte-inspired optical systems with distinct tunable reflectance according to a series of internal and external factors. More recently, scientists have made creative attempts to engineer mammalian cells to explore the function potentials of reflectin proteins as well as their working mechanism in the cellular environment. Progress in wide scientific areas (biophysics, genomics, gene editing, etc.) brings in new opportunities to better understand reflectins and new approaches to fully utilize them. The work introduced the composition features, biochemical properties, the latest developments, future considerations of reflectins, and their inspiration applications to give newcomers a comprehensive understanding and mutually exchanged knowledge from different communities (e.g., biology and material).

## 1. Introduction

The capability to rapidly control skin colors is rare, but pigment or structural discoloration is still fantastic in the animal kingdom [1,2,3]. However, cephalopods (squid, octopuses, and cuttlefish), as masters of disguise, can dynamically morph into or appear from the ambient environment by tuning the pigmentation, iridescence, transparency, and light-scattering of their skins [4,5,6,7,8,9]. Their hallmark ability to rapidly change overall body coloration inspires scientists, engineers, and hobbyists since being recorded by Aristotle in 350 BCE in his famous *Historia animalium* [10]. 

The cellular basis for cephalopods to produce unparalleled structural and pigmentary colors has been gradually understood after decades of research. Expandable chromatophore organs selectively absorb light by the reversible extracting and stretching of pigmentary sacs in the uppermost layers of the dermis. Thus, skin color is regulated by the area proportion among different chromatophores [11,12]. On the other hand, two kinds of specific cells, called iridocytes and leucophores, are responsible for producing structural coloration. Iridophores, with the cell biology of reflective cells, present iridescence through multi-layer interference [4,13], while the bright white color is the result of unselective light scattering by granular vesicles in leucophores [7,14], as well as by multilayer reflectors (Bragg stacks) in a few Cephalopods [15]. The latest research results show that the reflectin-containing sheath cells enveloping the pigment sac can produce structural colors without iridophore-like plates [7]. While our understanding of cephalopods’ color changes is still growing, these three types of cells synergistically display nearly every wavelength of visible light (see Figure 1) [16]. Rich color rendering can be used for camouflage and allow animals to exchange information or recruit similar species when they are cruising in their natural habitats [17]. 

Recently, scientists have been particularly interested in understanding the biological mechanism underlying squids’ dynamic iridescence produced by constructive interference. Cephalopods have used controllable iridescence to camouflage and communicate underwater for millions of years. Scientists, admiring their powerful color-change ability, worked hard to identify functional proteins in reflective tissues and explored the biochemical and functional properties of functional proteins [13,18,19,20,21,22]. Based on fundamental research, proteins called reflectins have been found and are used to fabricate myriad materials with distinct stimuli-responsiveness or even optical performances [5,23,24,25].

In a biochemical scope, Dr. Morse and his colleagues have accomplished brilliant works and reviewed relevant basic studies [14]. From the perspective of bio-inspired applications, Chatterjee et al. sorted out efforts to apply reflectins as materials to establish color-changing coatings and devices [26], which is also briefly mentioned by some other reviews [27,28,29,30]. However, the relationship between the biological prototypes and these man-made systems is poorly considered. As Morse and Dr. Levenson suggested, the material community sometimes overstates the role of functional proteins in iridocytes but overlooks other equally important components (e.g., membrane envelope) [14,22]. Therefore, the contours of squid dynamic iridescence should be sketched by introducing its material basis, relevant basic studies, and bio-inspired applications to bridge the knowledge of different areas. 

Based on this consideration, this mini-review comprehensively introduces reflectins, including what they are, how they are studied, and how they can be utilized (see Figure 1). Research begins with the discovery of a class of proteins called reflectins with a unique amino acid composition and ultra-high index (1.591 ± 0.002) [18], which lays the foundation for responsive optical functions of Bragg lamellae in iridocytes. Afterward, efforts are devoted to understanding the underlying mechanism of reflectin-tunable biochemical properties and sensitivity to environmental stimuli, as well as their collaborative mechanism with ACh-regulated phosphorylation effects and associated membrane systems [19,31,32]. Based on their unique material compositions and responsive self-organization characteristics, many thin films, fibers, and diffraction gratings are fabricated [5,23,33] to establish biophotonic structures with optical properties or photon-conductive abilities [34,35]. Lastly, most updated works revealed that reflectins can be used as novel molecular tools for synthetic biology, which establishes striking optical structures and achieves selective subcellular localization in mammalian cells [36,37]. It is an emerging and interesting research direction of reflectins but has not been adequately reviewed before. In short, reflectins, as unique biomaterials found in exotic squids, deserve to be more intensively studied and are likely to open up exciting application scenarios.

**Figure 1 ijms-23-15679-f001:**
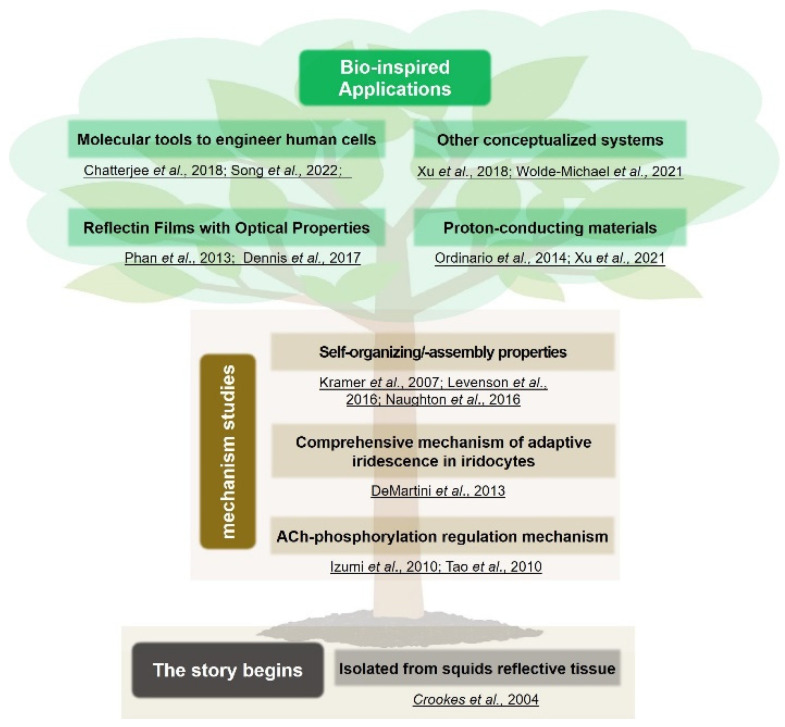
Timeline and roadmap for reflectin-associated studies, including their initial component identification, fundamental studies about their working mechanism, and novel inspiring bio-inspired applications [5,13,18,19,25,26,32,33,34,35,36,38,39,40,41]. Copyright 2007, Nature Publishing Group. Copyright 2009, Elsevier Ltd. Copyright 2004, Science Publishing Group. Copyright 2009, The Royal Society. Copyright 2018, Science Publishing Group. Copyright 2018, IOP Publishing Ltd. Copyright 2013, National Academy of Sciences of the United States of America. Copyright 2013, John Wiley & Sons, Inc. Copyright 2014, Nature Publishing Group. Copyright 2021, the American Chemical Society. Copyright 2022, Frontiers Publishing Group. Copyright 2016, the American Society for Biochemistry and Molecular Biology. Copyright 2017, AIP Publishing LLC. Copyright 2021, Nature Publishing Group. Copyright 2016, John Wiley & Sons, Inc.

## 2. Reflectins and Their Composition Features

The colorful iridescence of the Cephalopod is produced by a group of cells called iridophores, which were first thought to be static structures in contrast to expandable and extractable chromatophores. Iridophores in most species are static reflectors indeed, and ambient light instead of the animal determines its optical appearance [42]. Notable exceptions are found in the family *Loliginidae* [4,13]. The reflective states of tunable iridocytes in this family, including Atlantic (*Doryteuthis pealeii*) and Pacific (*Doryteuthis opalescens*) squids, can be reversibly and progressively activated by neurotransmitters, acetylcholine (ACh) [14,19,31,42,43]. 

The emergence of iridescent cells and associated mechanisms are supposed to be the result of remarkable species competition. Molecular evidence suggests that the predator–prey arms races shaped the body plan and diversification of modern cephalopods during the Mesozoic marine revolution [44,45]. Mighty predators and niche competitors spurred slow ancient cephalopods to abandon their heavy shells and acquire competitive mobility [46,47]. As they dominate nearly the whole ocean and flow into nearshores, optically dynamic environments urged loliginids (estimated ~150 million years ago) to evolve tunable dermal iridescence [14,48]. As a behavioral strategy, the ventral reflector directs light into the surrounding seawater to mimic down-welling moonlight and starlight. It can puzzle predators and prevent attacks from below [13,49,50]. Meanwhile, the sexual dimorphism of iridescence in squids, including features (location and tenability), is likely to serve as significant signals during mating or reproduction [4].

Other famous and glorious reflective structures can also be found in bird feathers, butterfly wings, and fish scales in the animal kingdom [51,52,53]. These micro-nanostructures of comparable sizes to the light wavelength are often composed of hard materials with high refractive indices such as purine crystals, keratin, or chitin. In contrast, the photon-active nanostructure of cephalopod reflective cells includes reflective proteins (soft protein substances) [14]. These reflectin proteins were first identified by Crookes et al. in squids’ reflective tissues of Hawaiian bobtail squid *Eupyrma scolope* [18] and exist in a variety of squid species, e.g., pelagic Pacific (*Doryteuthis opalescens*) and Atlantic (*Doryteuthis pealeii*) [4,19,20]. They are exclusively expressed by squids, with no homologs found in other species. These water-insoluble reflectin proteins are separated in the supernatant of an SDS-solubilized pellet and occupy ~40% of the whole proteinaceous component of the reflective tissues [18]. The reflectins (as soft optically photonic materials) may have the highest refractive index for natural proteins [5]. 

The unique properties (e.g., solubleness, self-assembly, and optical features) of reflectin are dominated by their unusual compositions. The family does not have common lysine, leucine, isoleucine, and alanine. However, the content of relatively rare tryptophan, arginine, methionine, and tyrosine is unconventionally high [18,19,20,54]. The peculiar proportion of aromatic and methionine residues is speculated to provide the π–π stacking interactions to drive reflectin assembly, especially when there is a lack of other hydrophobic residues [14].

Another feature of most reflectins is that they are protein-based block copolymers. Two 25-amino acid methionine-rich motifs are presented in the reflectin sequence: the N-terminal motif (RM_N_) [MEPMSRM(T/S)MDF(H/Q)GR(Y/L)-(I/M)DS(M/Q)(G/D)R(I/M)VDP(R/G)] [19] and a series of conserved reflectin motifs (RMs) [M/FD(X)_5_MD(X)_5_MDX_3/4_] [18]. The “X” sites represent amino acids enriched in R, G, D, S, and Q and depleted in Y, N, W, P, and F relative to the whole protein for RMs, while the subscript indicates the number of amino acids [14]. Compared with RMs, the N-terminal region is more conserved in reflectin isomers and cross-species evolution [14,19,38]. Positive-charged polyelectrolyte linker regions are interspersed between canonical reflectin motifs and have many arginine residues and aromas instead of negatively charged residues [14,34,38,39]. The evolvement of reflectins’ block structure and conserved reflectin motifs has been recently depicted, with evidence provided by the symbiotic bacteria *Vibrio fischeri* [55]. Guan et al. reported that 24 aa reflectin motifs can be further divided into three octapeptides [56]. These octapeptides can be traced back to *V. fischeri* symbiotic bioluminescent bacteria’ transposon [56]. Long time symbiotic process leads to the genetic drift of the transposon from *V. fischeri* to its squid host. Typical motifs and block structures of reflex proteins have been produced after the translocation and self-replication of transposons for millions of years [36].

## 3. Phosphorylation of Reflectin and Tunable Iridescence

The iridescence is static in the species *Euprymna scolopes*, where reflectin proteins were originally identified [14,19]. In the family *Loliginidae*, however, the neurotransmitter acetylcholine (ACh) activates reflectin platelets, and tunable reflective structures and dynamic iridescence are generated under the control of the muscarinic cholinergic system in the family *Loliginidae* [6,42]. This endogenous, stimulus-induced response is speculated to be directly related to the phosphorylation and dephosphorylation of reflectins [19]. Tao et al. have successfully observed the blueshift of iridophore cells by Ach stimulation and confirmed the progressive phosphorylation of RA1 during this process by 2D electrophoresis [13].

Izumi et al. suggested three possible mechanisms accounting for this phosphorylation-associated iridescence regulation [19]. (1) Reflectin’s net charge is changed by phosphorylation. It alters reflectin-containing platelets’ spacing, thickness, and/or refractive index and reflectins’ intermolecular interaction and assembly [14,38]. (2) The phosphorylation of reflectins alters the Gibbs–Donnan equilibrium in the iridocytes membrane, which influences variables such as the thickness, swelling pressure, protein migration, or protein aggregation of the membrane-bound structural reflectors [13,14,32]. (3) The phosphorylation of reflectins shifts assembly kinetics toward self-aggregation and hierarchical self-assembly by regulating its affinity for cell membranes [19,43]. Separately or synergistically, these three mechanisms lead to the observed changes in skin reflectance. 

Morse and coworkers have explored the biochemical properties of reflectins in vitro, and creatively reproduced the effects of phosphorylation by pH titration assays. As a component surrogate for phosphorylation, pH titration was used to control proteins’ electric charge, which achieve layered assembly and the reversible condensation of purified recombinant reflectins of *D. opalescens* exquisitely controlled by charge neutralization [14,38] (see Figure 2).

Morse and coworkers designed an ingenious experiment to directly record the reversible water flux in response to the neurotransmitter ACh [32]. The result delineates the whole picture and sorts the relationship among neurotransmitters, phosphorylation, reflectins, and structural coloration. Neurotransmitter ACh is released and binds to muscarinic receptors at the start [57]. Subsequently, a signal transduction cascade is triggered and activates the phosphorylation of reflectin proteins correlated with their condensation [19,32]. Lastly, D_2_O transport analysis demonstrates that the reversible condensation of reflectins drives Gibbs–Donnan-mediated water efflux through the lamellar cell membrane, which rapidly and reversibly dehydrates the lamellar membrane [32]. As the result, the multilayer reflector’s refractive and periodicity index is changed [20,32] (see Figure 3). The dynamic and controllable assembly property of reflectins is present in the scenario. 

## 4. Bio-Inspired Systems Based on Reflectins or/and Bragg lamellae Structures

The reversible and supramolecular assembly of reflectin proteins is the key mechanism in the facilitation of iridocytes’ dynamic optical function from a material perspective [13,20]. Hence, material science has paid attention to the Bragg lamellae structure of iridophores (as a structural interference device) or reflectin proteins (as a structurally adaptive biomolecule) to develop new, bio-inspired, adaptive optical devices [4,5,15,32].

Reflectins have been used for creating functional materials just three years after identification [18]. Kramer et al. reported the self-organizing of reflectins and the construction of protein-based thin films, photonic grating structures, and fibers. Specifically, the thin-film interference of reflectin films presents a broad-spectrum (λ = 400–800 nm) structural color responsive to the relative humidity of the environment [5]. Henceforth, increasing attempts are made to take full advantage of the distinct material properties of reflectins. Gorodetsky et al. used the water-soluble reflectin isoform of *Loligo (Doryteuthis) pealeii* to cast thin films on graphene oxide-coated substrates and examined their response to external stimuli [33]. Acetic acid vapor regulates the reflectance of protein coatings. It tunes across a wide spectral region of λ = 400–1200 nm (see Figure 4A), which is much larger than Kramer’s measuring range [5]. The modulation of surface reflectivity spectra is speculated to be induced by film expansion and changes in electrostatic interactions within proteins (see Figure 4B). Since the infrared range is covered, the visibility of the reflectin-based thin film is reversible under an infrared imaging camera, which shows its potential for thermal camouflage applications [23,24,29,33]. 

Crookes-Goodson et al. created a recombinant peptide refCBA derived from reflectin 1a of Hawaiian bobtail squid (*Euprymna scolopes*), which consists of the intervening C domain (DYYGRFNDYDRYYGRSMF) and the B (NYGWMMDGDRYNRYNRWMDYPERY) and A subdomains of repeat 2 (MDMSGYQMDMSGRWMD MQGR). Films based on these peptides display wide structural colors with wide ranges because of the reflected light interference of thin films, which is dominated by peptide concentration, lamellar stacks, and relative humidity [23] (see Figure 5). Similarly, they found a 23 amino acid region (DPRYYDYYGRFNDYDRYYGRSMF) in reflectin 1b, which is sufficient to reproduce the light scattering properties in the same way as full-length proteins [39]. These brilliant studies present that the motifs and linkers of reflectins can be separately or programmatically used. It is beneficial for both cost control and application expansion.

The above-mentioned examples show that reflectin-derived peptides and purified reflectins are utilized to form stunning arrays of diffracting gratings, fibers, and films. Its reflectance can be shrunk and adjusted according to changes in humidity, etc., in the above-mentioned examples [5,23,29,33]. However, Levenson et al. stated that errors keep recurring in the references that mistakenly regard reflectins per se as the whole photonic system in iridocytes [14]. On the one hand, the reflectin family is not the only composition of iridocytes Bragg reflectors. This integrated system is composed of alternating layers of low-refractive-index extracellular space (~1.36) and a high-refractive-index protein (~1.591). On the other hand, reflectins have dual functions: they work as a biomaterial with a high refractive index, as well as a motor that transfers molecular interactions into relevant mechanical forces and controls the Bragg lamellae’s rehydration and reversible dehydration. Reflectin functions can only be partially fulfilled without the multilayered structure. 

Emmanuel et al., cast multiple thin films of reflectin isoforms on Si wafers by spin-coating. These thin films occupy the incidence angle varying at 20 and 70° across a wide wavelength range of 185~3300 nm. Next, spin-coated BSA films are added as surrogates of the extracellular space (lower refractive index) to integrate with reflectin films, which restores the multilayered structure of cephalopods’ reflectin-based reflectors. The BSA layer can be further replaced by phytochrome to ulteriorly endow the system with dynamic tunability, which allows the subtle shift of the visible-light-induced isomerization of the optical properties of thin films [40] (see Figure 6) Although the preparation process of reflectin films is similar to that of the earlier systems, the reconstruction of multilayered reflectors with dynamic reflectance is highly original and creative. 

More conceptually, Gorodetsky et al. abandoned the employment of reflectins and developed a TiO_2_/SiO_2_ Bragg stack–modified device with low and high indices of refraction. The surface morphology of the bio-inspired multilayered structure can be mechanically actuated by adjacent integrated proton-conducting electrodes and a dielectric elastomer membrane, which regulates incident infrared reflectance and tunes its appearance properties under infrared-camera observations [25] (see Figure 7A). Likewise, beyond the color-change fields, the bulk protonic conductivity of reflectins and reflectin-derived polypeptides is also discovered, which is a pioneered attempt at developing protein-based protonic transistors [34,35] (see Figure 7B). Although these transistors lack correlations with color changes, the material composition of reflectins, for example, its polar charged amino acid side chains, plays a crucial role during proton transportation. The current density drops from 8.2 (±5.9) × 10^−2^ to 0.9 (±0.2) × 10^−2^ A cm^−2^ at 1.5 V by alanine (termed DE → A) to substitute glutamic acid and aspartic acid residues [34]. These outstanding studies elucidate the bio-derived semiconductor, which is a potential human-toxic and low-ecological electronic alternative for unsustainable electronics used today [58]. 

However, these delicate man-made systems still lag behind biological systems, especially in optical performance. They are very different from their native templates, for example, iridocytes or Bragg lamellae, in structure, morphology, or mechanism. At least two elements are absent from the currently developed bio-inspired systems, namely, the cytomembrane and cytoskeleton. The cytomembrane is indispensable for the Bragg lamellae in iridocytes enveloping reflectin platelets and separating the latter with the low-reflective-index extracellular space. On the other hand, the cytoskeleton is also essential for the transportation of certain cellular components and the maintenance of Bragg lamellae spatial conformation. From this dimension, it may be indispensable to either comprehensively investigate reflectins’ intracellular behaviors or take live cells as platforms for reflectins to fully carry their functions before we can develop high-performance bio-inspired photonic systems. 

## 5. Exploring Reflectins’ Functions and Behaviors in Cells

Recently, two different groups employed embryonic kidney (HEK-293) cells to explore the intracellular properties of reflectins. Tunable light scattering ability and transparency changes are given to engineering human cells by protein photonic structures [37,59]. Since one of these studies is published in a preprinted version and not formally issued [59], only Gorodetsky and coworkers’ publication is presented here [37].

The transfection of a vector encoding for RA1 realizes reflectins expression in large amounts (see Figure 8). The abundant foreign proteins then assemble into leucosome-like cytoplasmic aggregates that are widely distributed throughout the cell’s interiors, with unusually high refractive indices. Having been given sufficient time to express RA1, the phase difference between the cell bodies and the glass substrate becomes significant (see Figure 8B). More interestingly, RA1-expressing cells become more dark-gray and distinct from the environment upon high NaCl concentration (217 mM) exposure (see Figure 8C), as one result of reconfiguring reflectin-based subcellular structures. 

The work is a milestone in modifying mammalian cells with novel characteristics by discovering new molecular tools, which provides new concepts for biophotonic technologies [30]. However, many questions remain unanswered. Why does RA1 form leucosome-like aggregates or droplets in mammalian cells instead of Bragg lamellae? What is missing but indispensable? Moreover, RA1-engineered cells have slightly rounded morphologies compared with ordinary HEK-293T cells, but the researchers gave little explanation for this phenomenon.

Similarly, Song et al. realized the expression of the family *Loliginidae* reflectins A1, A2, B1, and C in mammalian cells. All reflectins are observed to spontaneously form aggregates but with distinct cyto-/nucleoplasmic localization preferences. To our knowledge, this is the first report that presents the differences among reflectins in cells, not by in-tube assays. Moreover, the cytoskeleton interacts extensively with protein, because RA1 is significantly enriched in the microtubule organization centers (see Figure 9). The cytoskeleton system is important in the transportation and organization of reflectin proteins, which further determines the morphology of Bragg lamellae [36]. The interplay between reflectins and the cytoskeleton system may also be the reason for the rounded morphology of RA1-engineered cells in Gorodetsky’s study [37].

Meanwhile, the distinct sub-cellular localization may be evidence to decipher the reflectins interaction with the membrane system. Song et al. reported the binding of RA1 and RC to the DOPC/DOPG phospholipid membrane, with different behaviors [60]. Additionally, DeMartini et al. found that RC is localized at or near the lamellar membrane, by immunolocalization experiments, while RA1 and RA2 are speculated to be responsible for the formation of the inside hydrogel network [20]. Their different affinities or other associated properties to cell membranes inevitably influence their subcellular localization observed by Song et al. [36], though the specific mechanism needs to be determined by further studies. 

Predictably, it will be interesting to explore reflectins’ interaction with intracellular components (e.g., cytoskeleton and cytomembrane). These studies will clarify the working mechanisms of reflectins in the dimension of basic research and also enhance our capability to construct the intracellular photonic system. As a suggestion and a plan for what we are working on, cephalopod skin cells or polarized cells with highly organized cytoskeletons can be employed as platforms to establish more delicate and effective reflectin-based photonic structures since their sophisticated cytoskeleton systems will provide reflectins with effective transportation and distribution networks. Meanwhile, maneuverability can be further improved with the developed gene sequencing/assembling technology and novel gene editing methods. Recently, high-quality genomes of both *Sepia pharaonic* and *Doryteuthis pealeii* are assembled, which paves the way to genetically manipulate cephalopods cells and better understand the mechanism underlying cephalopods’ unique features [61,62]. For example, Crawford et al. used the CRISPR-Cas9 system to realize the knockout of Tryptophan 2,3 Dioxygenase genes in the squid *Doryteuthis pealeii*, which eliminates the squid embryos’ pigmentation [63]. Similarly, since genomic information is now available, it is feasible to precisely knock out reflectin genes in squid embryos. It brings great chances to observe the function of reflectins during establishing those exotic biophotonic structures in iridocytes. 

Moreover, albeit not for photonic applications, the construction of reflectin assemblies in cells can establish models for wider research fields. As Atrouli et al. suggested in their enlightening review, the dynamic self-assembly properties or tendency to form aggregates in vitro causes the potential surrogates of reflectins to simulate the pathogenesis of neurological disorders, because these diseases are supposed to be caused by the formation of misfolded protein aggregates [27]. Now that we can construct intracellular reflectin-based aggregates in cells, we may better understand how these misfolded proteins trigger neurological diseases under physiological conditions. 

## 6. Conclusions and Perspectives

Reflectins have unusual amino-acid compositions and sequential properties, which endow themselves with properties including an extremely high reflective index (1.591 ± 0.002), and responsiveness to various environmental stimuli [18]. Based on their unique material compositions and self-organization properties, reflectin-derived peptides and purified reflectins are fabricated into an impressive array of thin films, fibers, diffraction gratings, and complex multi-layered optical systems [5,23,25,29,33,40]. More recently, mammalian cells have been creatively used as platforms to explore the function potentials of reflectin proteins and investigate their working mechanisms in the cellular environment [36,37].

Nevertheless, these delicate man-made devices still lag behind biological systems (e.g., iridocytes) in optical performance, because biology has been used for specific applications (e.g., coloration, camouflage, and signaling) for over 100 million years [16,64]. Human beings can build up space shuttles or mansions, but it is so far impossible to construct one seemingly ordinary cell de novo.

To our understanding, there are at least three promising approaches towards high-performance reflectin-based photonic technologies. (1) It is necessary to understand the design of biological materials or biological materials with dynamic and self-regulating functions from a synthetic perspective. Basic research should be carried out to better understand reflectins’ structure, self-assembly, and their tendency to form aggregates under physiological conditions [27,65]. (2) As membrane envelopes serve as amplifiers to the range of tunable sizes of reflectin assemblies [22], it is necessary to decipher the mechanism of how reflectins interact with cytomembranes. This is also an interesting but puzzling issue because reflectins are verified to interact with lipid membranes [60] but without likely transmembrane, alpha-helix, or beta-sheet regions [19]. (3) Last but not least, living cells can be used as valuable platforms to explore reflectins’ working mechanisms and fully exhibit their functions because cells can provide reflectins with sophisticated components (e.g., cytoskeletons, cytomembranes, and dynamic physiological stimuli) for reflectins to construct intracellular photonic structures [36,37], which is absent in any man-made system so far.

## Figures and Tables

**Figure 2 ijms-23-15679-f002:**
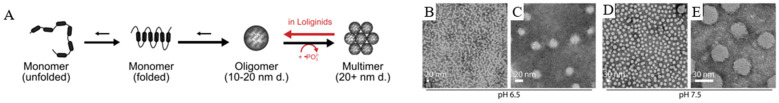
(**A**) Net charge neutralized progressive and reversible hierarchical assembly driven by pH-titration (in vitro) or phosphorylation (in vivo) [14]. Copyright 2017, AIP Publishing LLC. (**B**–**E**) TEMs of reflectin A1 assemblies at pH6.5 and pH7.5 at low and high magnifications [38]. Copyright 2015, American Society for Biochemistry and Molecular Biology.

**Figure 3 ijms-23-15679-f003:**
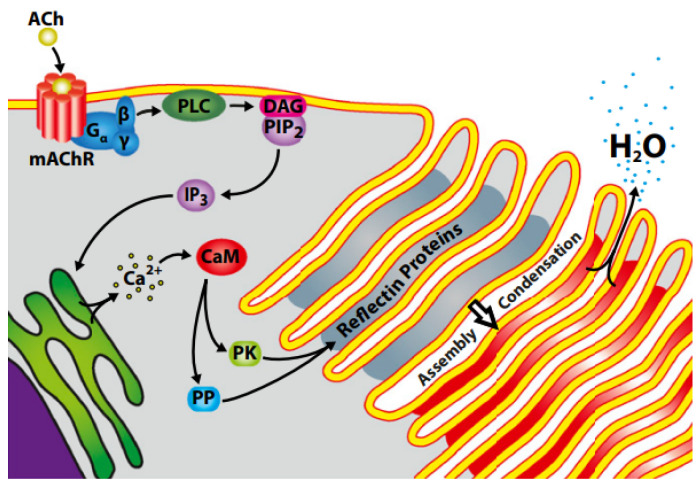
Proposed ACh-activated reflectin platelet condensation and iridescence tuning [32]. Copyright 2013, National Academy of Sciences of the United States of America.

**Figure 4 ijms-23-15679-f004:**
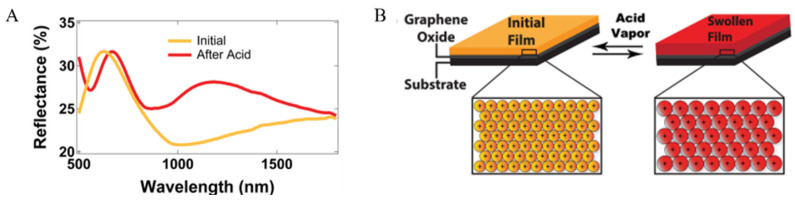
(**A**) The response changes of the reflection spectrum of the RA1 film layer before and after the acetic acid treatment; (**B**) Changes in the electrostatic state of RA1 nanoparticles before and after the acetic acid treatment and the expansion of the film layer [33]. Copyright 2013, WILEY-VCH Verlag.

**Figure 5 ijms-23-15679-f005:**
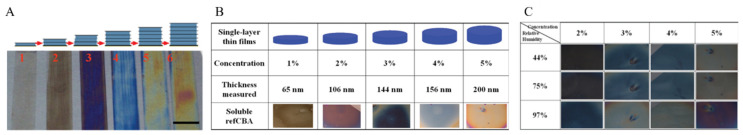
(**A**) RefCBA films of different layers (as red numbers in the figure) prepared by the spin-coating method; (**B**) RefCBA films of different thicknesses; (**C**) Dynamic color response of RefCBA films of different thicknesses under different relative humidity [23]. Copyright 2013, Wiley Periodicals, Inc.

**Figure 6 ijms-23-15679-f006:**
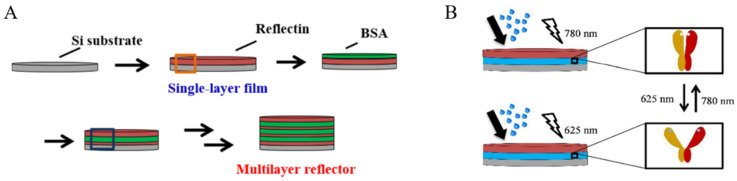
(**A**) Structure of the reflectin/BSA composite sheet layer inspired by cephalopods; (**B**) Light response of the phytochrome/reflectin composite sheet layer [40]. Copyright 2021, Nature Publishing Group.

**Figure 7 ijms-23-15679-f007:**
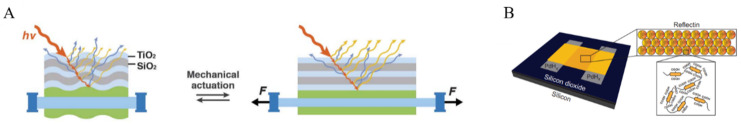
(**A**) Surface reflection before and after the mechanical stretching of the TiO_2_/SiO_2_ composite sheet layer [25]; Copyright 2018, Science Publishing Group. (**B**) Biological semiconductor based on the film layer of reflectins [34]. Copyright 2014, Nature Publishing Group.

**Figure 8 ijms-23-15679-f008:**
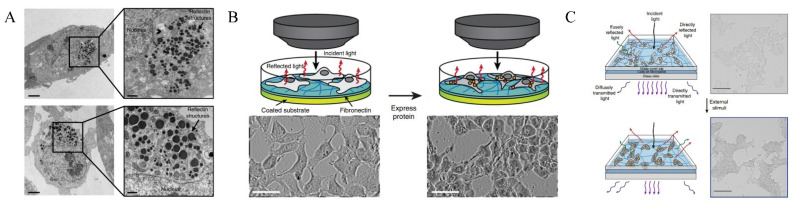
(**A**) Electron-dense structure of RA1 particles based on the cross-sectional view of the cells expressing RA1 observed by TEM; (**B**) Optical characteristics of the cells after expressing RA1 proteins (compared with the Petri dish substrate, the contrast increases); (**C**) changes in the transparency of cells expressing RA1 proteins before and after the treatment with sodium chloride solutions with the scale bar = 25 µm [37]. Copyright 2020, Nature Publishing Group.

**Figure 9 ijms-23-15679-f009:**
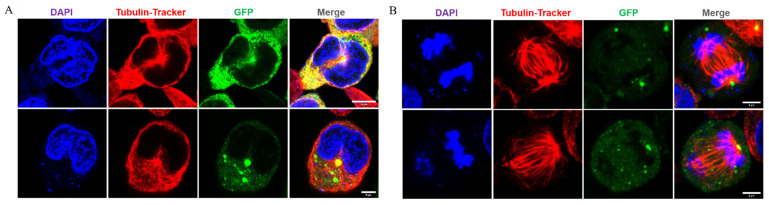
(**A**) Co-positioning of reflectins and the cytoskeleton in the cell center, in interphase cells, with the scale bars = 10 μm. (**B**) Positioning of reflectins in the spindle center at the late stage of cell division, with the scale bars = 5 μm. Nuclei and microtubules are stained with DAPI (blue) and Tubulin-Tracker Red (red); RA1 is indicated by tandem EGFP (green); Yellow comes from the merging of green and red [36]. Copyright 2022, Frontiers Publishing Group.

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
