# Peer review of "A Mini-Review on Reflectins, from Biochemical Properties to Bio-Inspired Applications"

_ijms, 2022, doi:10.3390/ijms232415679_

Round 1
Reviewer 1 Report
Junyi Song and co-authors have prepared a review titled “A Mini Review on Reflectins, from Biochemical Properties to Bionic Applications”. The review discusses a protein from squid called reflectin, its phosphorylation, some work with thin films made of the protein, and expression of the protein in cells. Despite the potential for an interesting review given the authors’ combined knowledge and experience in the field, the manuscript in its current state is redundant, lacks focus, poorly referenced, and lacks new insight or commentary on the field. As a result, significant edits are required to make this manuscript ready for acceptance. Additional comments for major revisions are below:
· The authors should clearly state the focus of the review and its significance. There are many reviews on reflectins, what makes this review different from the others? Although lines 27-29 provide an outline for this manuscript, it is not accurate to what is actually written. For example, no critical thinking on the future development of reflectins is provided. Also, no bionic applications are discussed (note, bionic by definition means: having normal biological capability or performance enhanced by or as if by electronic or electromechanical devices per the Merriam-Webster dictionary).
· Many references are missing or are incorrectly cited. The references should be redone from scratch. Some examples are below:
o Line 35 – there are many references on structural or pigmentary coloration in nature; at least three general reviews and/or books should be cited
o Line 40 – the authors mention Aristotle, but do not cite the primary source
o The paragraph starting at line 67 provides various facts previously reported but no references are provided. Similar issues are present in lines 96-100, 104, 111-112, 158,192-196
o Several sentences have incorrect references, where the papers cited do not contain the information described in the sentence (see line 116,130, 135,137)
o Other sentences do not cite all relevant literature like lines 126,142, 173, 206,242
· Several paragraphs repeat portions of the manuscript and/or the abstract (lines 216-226, 378-410). Rather than repeat what has already been done with reflectins, the authors should focus on providing new insight into the findings or bringing forward discussion points that separate their work from existing reviews in the field. For example, they could elaborate on lines 270-275 – what is the relevance of reflectins in the optical structures they form? If other biological materials or structures could be more useful for further research, the authors should provide this information.
· Song and authors provide new information in the conclusion (418-424 and 431-436) regarding the genomics related studies. 1) this information should not be in the conclusion 2) elaborating on the genomics, which seems to be a specialty of the authors could be a new way of dissecting the function, behavior, and role of reflectins, which would give new insight into the protein and could be a great focal point of the review. The authors should consider realigning their manuscript to highlight the genomic aspects. The author should then rewrite the conclusion to provide a concise summary of the manuscript followed by a descriptive analysis of the future directions for this work rather than a long regurgitation of portions of the manuscript followed by new information.
Minor comments:
- The authors should spell-check their manuscript
- The authors should provide clear transitions to enhance the readability of their paper
- The authors should check their references and avoid citing documents that are not peer-reviewed, see reference in line 372
Author Response
Dear reviewer,
Thanks for your valuable suggestions. We have made the modifications throughout the whole manuscript according to your comments.
We have added new statements at the last of introduction part, to better present the purpose of this mini-review.
We have added relevant refs in 35, 40, 67-82, 104, 111-112, 116, 126, 130, 135, 142, 158, 173, 206, 242 as you suggested. However, the line numbers have been changed since the modifications.
For line 96-100, the magnificent evolution process was described in Ref. 10&30, which have been cited at the end of the paragraph.
For line 192-196, the statement was quoted from Ref. 15&36, which has been cited at the end of the paragraph.
For line 135 “The N-terminal region is more evolutionarily conserved across species and reflectin isoforms (23 in 27 kinds of the most known reflectins) than canonical RMN” has been rewrite as “Compared with typical RMN [37], the N-terminal region is more conserved in reflectin isomers and cross-species evolution. Very few alternative residues usually only appear in one or a few reflectin motifs [38], although a specific residue makes up almost all “X” sites”. We mistakenly quoted the former sentence from our earlier draft.
Besides, for those not peer-reviewed references, we either deleted them or keep them in the manuscript with explains. For example, “Since one of these studies is published in preprinted version and not formally issued [50], only Gorodetsky and coworkers' publication is presented here [49]”
As to the genomics related studies, we have brought them forward to a new part “5. Exploring reflectins’ functions and behaviors in cells”, and added some new perspectives for the future studies according to our understandings.
At last, we reorganized the “6. Conclusions and Perspectives” according to your instructive suggestions, by briefly summarized the works that have been done and more directly present our recommendations (even some plans) for the future works.
Thanks again for your comments.
Looking forward to your further suggestions.
Best,
-Junyi

Reviewer 2 Report
Review of Manuscript for International Journal of Molecular Sciences: A Mini Review on Reflectins, from Biochemical Properties to Bionic Applications.
Manuscript ID: ijms-1955513
Recommendation: minor revision
In this review article, the authors aim to compile information on the composition features, biochemical properties, state-of-the-art progress of reflectins and associated bionic applications.
The subject of the review is potentially interesting. However, the manuscript needs language editing by a professional before being considered for publication.
Several grammar, sentence structure, word choice, and spelling mistakes are found in the text.
For example:
Line 28: Overall, the work introduced..- Maybe the work introduces or compiles..?
Line 41: The cellular basis is recognized to generate its extraordinary structural and pigmentary colors after decades of research. -The sentence is awkward and needs to be rephrased.
Line 47: cell-biology meaning..- meaning..?
Line 49: that cephalopods expanded chromatophores.. -The sentence is awkward
Line 53: Not only for protection purposes, but the rich coloration features are also an important capability that enables communication or recruitment as these animals navigate their natural environments. -The sentence is awkward and needs to be rephrased.
Line 65: now human beings are looking to them..- Awkward phrase..
Line 81: are deserved..- Perhaps “deserve” sounds better.
Line 98: predator predators..
Line 123: Several common residues, saying..- Several common residues, namely..
Line 148: Guan et al. reported that 24 aa reflectin motifs can be further divided into three octapeptides in 2020. – the “in 2020” is not necessary and can be removed-the same goes to similar parts within the text.
Line 223: the material science community is looked to..The sentence is awkward and needs to be rephrased.
Line 272: molecular machine- molecular mechanism
Line 404: in another dimension..- in another approach perhaps sounds better
Moreover, there is already a recent review by Chatterjee. (Chatterjee, A. At the Intersection of Natural Structural Coloration and Bioengineering. Biomimetics 2022, 7, 66. https://doi.org/10.3390/biomimetics7020066) concerning reflectins and their potential applications. This review has escaped the attention of the authors and it has not been cited in this manuscript.
The references should be formatted according to the Journal’s guidelines. (refs 17, 20 and 23 are missing Journal name)
Overall, the manuscript ijms-1955513 needs improvement to meet the quality criteria for consideration as a contribution to the journal Journal of Molecular Sciences.
Author Response
Dear reviewer,
Thanks for your valuable suggestions. We have made the modifications according to your comments.
One editing company has been employed to polish the whole manuscript. We have left the modification trace so that you can easily find them.
More specifically, we have made modifications including but not limited to line 28, 41, 47, 49, 53, 65, 81, 98, 123, 148, 223, 272, 404.
Besides, Chatterjee inspiring review has also been added in the revised manuscript.
As to the references format, we are really sorry for that. The software Endnote sometimes makes mistakes like that. We have manually modified them.
Thanks again for your comments.
Looking forward to your further suggestions.
Best,
-Junyi

Round 2
Reviewer 1 Report
Junyi Song and co-authors have prepared a review titled “A Mini Review on Reflectins, from Biochemical Properties to Bio-inspired Applications”. The review discusses a squid protein called reflectin their properties in squid cells, films, and cells. From their last draft, the authors appear to have made several changes to the text, which have primarily been flourishes of language rather than to improve the technical content of the manuscript. Many of the issues persist from the previous version of the manuscript and relate to overall redundancy of the manuscript, the lack of focus and direction of the manuscript, persistent referencing issues, and a lack on new insight or commentary on the field. As such, I still maintain that further significant edits are required before this manuscript can be considered ready for acceptance. Additional comments for major revisions are below:
· The authors do not explicitly state the novelty of their review (i.e., what sets it apart from several other reviews in the field). This statement should be included at the end of the abstract as well as at the end of the introduction. (See previous comment: The authors should clearly state the focus of the review and its significance. There are many reviews on reflectins, what makes this review different from the others? Although lines 27-29 provide an outline for this manuscript, it is not accurate to what is actually written. For example, no critical thinking on the future development of reflectins is provided.)
· Many references are still missing or incorrectly cited despite the fact that several of the issues were pointed out during the last iteration (see previous comment: Many references are missing or are incorrectly cited. The references should be redone from scratch. Some examples are below: Line 35 – there are many references on structural or pigmentary coloration in nature; at least three general reviews and/or books should be cited; Line 40 – the authors mention Aristotle, but do not cite the primary source; The paragraph starting at line 67 provides various facts previously reported but no references are provided. Similar issues are present in lines 96-100, 104, 111-112, 158,192-196; Several sentences have incorrect references, where the papers cited do not contain the information described in the sentence (see line 116,130, 135,137); Other sentences do not cite all relevant literature like lines 126,142, 173, 206,242). Examples of erroneous references are below:
o Line 41: Several key iridophore references from the Morse group and others are missing
o Line 43: Reference to Aristotle’s work is still missing
o Line 78: the study by Kramer et al is not cited despite using the refractive index determined from that work
o Line 98: missing and incorrect references
o Line 167: the work by Tao et al should be cited and discussed
o Line 174, 182, 253, and others: reference 10 (review) is overcited and should be replaced with citations of the original papers that describe the work.
· Several paragraphs repeat the same information (see previous comment: Several paragraphs repeat portions of the manuscript and/or the abstract (lines 216-226, 378-410). Rather than repeat what has already been done with reflectins, the authors should focus on providing new insight into the findings or bringing forward discussion points that separate their work from existing reviews in the field.). The repeated portions should be trimmed and replaced with insight/discussion from the authors regarding the field. Relevant areas where expansion would be of interest to the readers are below:
o The purpose of including chromatophores and leucophores is unclear given that the work seems to focus on iridophores based on the abstract. Either the discussion on chromatophores and leucophores should be removed or their relevance to the iridophore studies should be further expanded.
o Paragraph starting at line 60 – if the focus of the paper is on iridophores, this paragraph should be expanded to describe the capabilities of iridophores and their relevance to the different fields described like biochemistry and materials science
o Line 71-74: what are the mistakes and what is the correct mechanism? The authors should clearly lay out their opinion on this topic or provide a succinct summary from a previous review that clarifies the existing issue(s) in the field.
o Line 176: how does self-assembly by regulating its affinity for cell membranes, particularly with respect to the different isoforms?
o Line 194: correlation of Gibbs-Donnan mediated water efflux with proton transport observed in films would provide an interesting correlation between some of the self-assembly and functional studies performed for reflectins and would add to the novelty of this manuscript (otherwise, the section starting at line 291 seems irrelevant to the review and should be removed)
o Line 252-253: how can reflectin domains be Lego bricks? What are the synthetic biology applications? The authors should expand on these proposed utilities of reflectins either here or in the conclusion.
o Line 350-353: The localization of the isoforms in mammalian cells should be compared to what was found in iridophores by DeMartini et al.
· There are several instances of technical inaccuracies throughout the manuscript, which should be corrected. For example:
o Line 51-54: bright white color is also from iridophores for some cephalopods
o Line 51-54: retractable pigment cell does not have reflectin, the sheath cells that are separate from the pigment sac have reflectin
o Line 114: Reflectin homologues have been found in species other than squid. If the authors have found otherwise, they should provide the appropriate references and delineate the error in this manuscript.
o Line 115: the authors should clarify what organs contain 40% reflectin, keeping in mind that iridophores are cells not organs and that only one cell in the chromatophore organ contains reflectin, this value appears to be somewhat high. Additional details are necessary to present the number accurately.
o Line 127: Which reflectin isoform has 57% of the rare amino acids? The isoform and the relative variability (average/standard deviation or similar statistics) should be provided if discussing all known reflectin isoforms. Additional references are also required here.
o Line 135-138: The authors should state which reflectin motifs (line 136), and what residue (line 137) make up the repeated motifs. By clearly defining these amino acids, readers from other disciplines may also find commonalities between the reflectin work and their work broadening the readership of this manuscript.
o Line 170: Reflectins are not contained in ribosomes. If the authors have found otherwise, the relevant reference that directly states reflectins are contained within ribosomes should be cited.
o Line 173, 174: what is the iris-ciliary body membrane and the membrane-bound iris body? Where are they located and what references describe reflectin attachment to these bodies? Direct references should be used rather than a general review.
o Line 266 contradicts line 267: reflectin has a high refractive index, i.e., it interacts with light and therefore is a photonic material. The authors should more clearly define the associated molecular mechanism such that the photonic property of the protein can be distinguished from its function, which seems to be what the authors are trying to accomplish here, but is not clear from the statements currently presented in the work. The authors should also introduce this concept early on in the manuscript and analyze the studies presented within this context
o Line 328 the authors say they don’t discuss the preprint but it is discussed in the following page.
· The authors should completely rearrange their manuscript to provide information in a logical and relevant manner. For example:
o The authors should separate in vivo results from in vitro results from cell work. The intro duction and sections 1, 2, 3 mix these concepts making it difficult to fully evaluate the results.
o Line 99: the discussion of the light organ of E scolopes should be moved here.
o Line 116-122: is redundant and should be removed or condensed.
o Line 144-147: the relevance of this section to the subsequent section and to iridophores is unclear.
o The authors have several instances of single sentence paragraphs (see line 156-158), which make the text challenging to read. These “paragraphs” should be combined with the paragraphs before them to enhance the overall readability of the manuscript.
o Line 201-210 and 211-218 is redundant and unnecessary
o Should combine lines 227 and 228
o Transitions required before line 284, line 291
o Move 306-310 to conclusion
o The authors discuss various studies to different extents. Their rationale for deeply analyzing some studies and not others is not clear (see page 10 vs 6-8). Mammalian cell related work accounts for 2 studies out of all reflectin work whereas the film studies are the bulk of the in vitro work on reflectins. The goal of these analyses should be clarified and they should be discussed proportionally.
o Line 389-414 is redundant and should be removed from the conclusion.
o The authors introduce new concepts in line 433. This is unnecessary in the conclusion and should be discussed in the main body.
o Line 436-438 This statement does not conclude the text and is generally difficult to follow – what are polarized cells and what part of reflectin research discussed in this paper has any bearing on the field? Rather than focusing on applications from single studies, the authors should provide broader applications for the findings obtained from reflectin studies both in vivo and in vitro.
· The authors have 17 instances of “Refractive index” or “indices of refraction” in their text but they argue that reflectins are not biologically active photonic structures (line 266). Why do the authors emphasize the refractive index so much in the manuscript, in this case? This goes back to my first comment – the goal and novelty of this review is unclear. The authors should explicitly clarify the novelty and based on that determine whether the emphasis on refractive indices is necessary or should be reduced throughout.
Minor comments (repeated from previous review):
- 1) The authors should spell-check their manuscript
- 2) The authors should provide clear transitions to enhance the readability of their paper
- 3) The authors use “RfA1” but do not define it – the meaning of RfA1 should be defined in the main text or the caption at the first use.
Author Response
Dear reviewer,
Thanks so much for conscientiously and carefully reviewing our article. We admire your responsible attitude and your specialized knowledge on Cephalopods structural color-changing.
Since there are so many citations need to be corrected, we highlighted them in the PDF-version.
For the rest, the replies are listed as follow:
Comment: Line 41: Several key iridophore references from the Morse group and others are missing
Reply: “However, cephalopods (squid, octopuses, and cuttlefish), as masters of disguise, can dynamically morph into or ap-pear from the ambient environment by tuning pigmentation, iridescence, transparency, and light-scattering of their skins”. In this sentence, no specific attention is given to iridophore. Brilliant works of Morse and others are cited at other paragraphs.
Comment: The purpose of including chromatophores and leucophores is unclear given that the work seems to focus on iridophores based on the abstract. Either the discussion on chromatophores and leucophores should be removed or their relevance to the iridophore studies should be further expanded.
Reply: Although this manuscript has little discussion on chromatophores or leucophores except for this paragraph, we think this brief introduction of 3 types of cells will be helpful for readers to know cellular basis of color-change of Cephalopods. It is also a transition paragraph to draw out structural color-change and iridocytes. However, if you insist, we are ok to simply remove this paragraph.
Comment: Paragraph starting at line 60 – if the focus of the paper is on iridophores, this paragraph should be expanded to describe the capabilities of iridophores and their relevance to the different fields described like biochemistry and materials science.
Reply: We have overally rewrited this paragraph to better draw out the purpose of this manuscript “Recently, scientists are particularly interested in understanding the biological mechanism underlying squids' dynamic iridescence produced by constructive interference. Cephalopods have used controllable iridescence to camouflage and communicate underwater for millions of years. Scientists, admiring their powerful color-change ability, worked hard to identify functional proteins in reflective tissues and explored the biochemical and functional properties of functional proteins. [13,17-21]. Based on fundamental researches, proteins called reflectins are found and used to fabricate myriad materials with distinct stimuli-responsiveness or even optical performance [5,22-24].”
Comment: Line 71-74: what are the mistakes and what is the correct mechanism? The authors should clearly lay out their opinion on this topic or provide a succinct summary from a previous review that clarifies the existing issue(s) in the field.
Reply: We have the expression to better present the idea: “As Morse and Dr. Levenson’s suggested, the material community sometimes overstates the role of functional proteins in iridocytes, but overlooks other equally important components (e.g., membrane envelope) [14,21]. Therefore, the contours of squid dynamic iridescence should be sketched by introducing its material basis, relevant basic studies and bio-inspired applications to bridge the knowledge of different areas.”
Comment: Line 176: how does self-assembly by regulating its affinity for cell membranes, particularly with respect to the different isoforms?
Reply: This is almost the original expression in the Ref. [18,38]. In addition, authors of Ref. [63] also observed that reflectin proteins can effectively bind and reorganize synthetic phospholipid vesicles. These works all indicates that reflectins occupy active membrane-associated properties, though the specific mechanism needs to be determined by further studies.
Comment: Line 194: correlation of Gibbs-Donnan mediated water efflux with proton transport observed in films would provide an interesting correlation between some of the self-assembly and functional studies performed for reflectins and would add to the novelty of this manuscript (otherwise, the section starting at line 291 seems irrelevant to the review and should be removed)
Reply: As we suggested in the manuscript, it is a conceptual work and its proton-conducting property is relevant with the amino acid composition of reflectin. So we think it is still relevant with reflectin. Besides, in Gorodetsky’s several works, we did not notice any description about Gibbs-Donnan. We haven't been able to connect them together yet. If you insist, we will be okay to remove the work “Bulk protonic conductivity in a cephalopod structural protein” from the manuscript.
Comment: Line 252-253: how can reflectin domains be Lego bricks? What are the synthetic biology applications? The authors should expand on these proposed utilities of reflectins either here or in the conclusion.
Reply: We have changed the sentences into “These brilliant studies present the motifs and linkers of reflectins can be separately or programmatically used. It is beneficial for both cost control and application expansion.” In the last round of revision, they have been left them by mistakes. We are sorry about that.
Comment: Line 350-353: The localization of the isoforms in mammalian cells should be compared to what was found in iridophores by DeMartini et al.
Reply: We added discussion about DeMartini’s work and rewrited the sentences: “Meanwhile, the distinct sub-cellular localization may be the evidence to decipher the reflectins interaction with the membrane system. Song et al., reported the binding of RA1 and RC to DOPC/DOPG phospholipid membrane, with different behaviors [63]. Also, DeMartini et al. found that RC is localized at or near lamellar membrane, by immunolocalization experiments, while RA1 and RA2 are speculated to be responsible for the formation of the in-side hydrogel network [19]. Their different affinities or other associated properties to cell membrane inevitably in-fluence their subcellular localization observed by Song et al.[35], though the specific mechanism needs to be determined by further studies.”
Comment: Line 51-54: bright white color is also from iridophores for some cephalopods.
Reply: Thanks for you reminding. We have added relevant Ref here.
Comment: Line 51-54: retractable pigment cell does not have reflectin, the sheath cells that are separate from the pigment sac have reflectin
Reply: Thanks for you reminding. We have corrected the statement “The latest research results show that the reflectin-containing sheath cells enveloping pigment sac can produce structural colors without iridophore-like plates [7].”
Comment: Line 114: Reflectin homologues have been found in species other than squid. If the authors have found otherwise, they should provide the appropriate references and delineate the error in this manuscript.
Reply: Dear reviewer, we are eager to know the reflectin-homologues expressed in other species. Would you please be more specific, what are these species?
Comment: Line 115: the authors should clarify what organs contain 40% reflectin, keeping in mind that iridophores are cells not organs and that only one cell in the chromatophore organ contains reflectin, this value appears to be somewhat high. Additional details are necessary to present the number accurately.
Reply: We have corrected “reflective organs” to “reflective tissues”, which should be more correspond to what is stated in reference “Reflectins: The Unusual Proteins of Squid Reflective Tissues”
Comment: Line 127: Which reflectin isoform has 57% of the rare amino acids? The isoform and the relative variability (average/standard deviation or similar statistics) should be provided if discussing all known reflectin isoforms. Additional references are also required here.
Reply: The proportion is provided by Ref. [18]: We have changed the statement and added more relevant references: “The family does not have common lysine, leucine, isoleucine, and alanine. However, the content of relatively rare tryptophan, arginine, methionine, and tyrosine is unconventionally high [18-20,50].”
Comment: Line 135-138: The authors should state which reflectin motifs (line 136), and what residue (line 137) make up the repeated motifs. By clearly defining these amino acids, readers from other disciplines may also find commonalities between the reflectin work and their work broadening the readership of this manuscript.
Reply: Dear reviewer, thanks for you suggestion. We have added one sentence to make clear the residues:” The “X” sites represent amino acids enriched in R, G, D, S and Q, and depleted in Y, N, W, P and F relative to the whole protein for RMs, while the subscript indicates the number of amino acids [51]. "
Comment: Line 170: Reflectins are not contained in ribosomes. If the authors have found otherwise, the relevant reference that directly states reflectins are contained within ribosomes should be cited.
Reply: We don’t know why there is a “ribosomes” here. We have deleted it. Sorry about this mistake.
Comment: Line 173, 174: what is the iris-ciliary body membrane and the membrane-bound iris body? Where are they located and what references describe reflectin attachment to these bodies? Direct references should be used rather than a general review.
Reply: The editing company changed the original “iridocytes” into “iris body”. We have corrected these mistakes. Thanks for you reminding.
Comment: Line 266 contradicts line 267: reflectin has a high refractive index, i.e., it interacts with light and therefore is a photonic material. The authors should more clearly define the associated molecular mechanism such that the photonic property of the protein can be distinguished from its function, which seems to be what the authors are trying to accomplish here, but is not clear from the statements currently presented in the work. The authors should also introduce this concept early on in the manuscript and analyze the studies presented within this context
Reply: We have changed the statement here, and in the introduction: “However, Levenson et al. stated that errors keep recurring in the references that mistakenly regard reflectins per se as the whole photonic system in iridocytes [14]. On the one hand, the reflectin family is not the only composition of iridocytes Bragg reflectors. This integrated system is composed of alternating layers of low-refractive-index extra-cellular space (~1.36) and a high-refractive-index protein (~1.591). On the other hand, reflectins have dual func-tions: they work as a biomaterial with high refractive index, as well as a motor that transfers molecular interactions into relevant mechanical forces and controls the Bragg lamellae’s rehydration and reversible dehydration. Reflectin functions can only be partially fulfilled without the multilayered structure.”
Comment: Line 328 the authors say they don’t discuss the preprint but it is discussed in the following page.
Reply: The preprint is only cited in this line. We have not discussed it at other places.
Comment: The authors should separate in vivo results from in vitro results from cell work. The introduction and sections 1, 2, 3 mix these concepts making it difficult to fully evaluate the results.
Reply: We have reorganized the introduction and make in vivo results in a separated part : 5. Exploring reflectins’ functions and behaviors in cells
Comment: Line 99: the discussion of the light organ of E scolopes should be moved here.
Reply: We have moved discussion of potential behavioral strategy of squids structural dynamic iridescence here. Not sure if this is what you suggested. ” The emergence of iridescent cells and associated mechanisms are supposed to be the result of remarkable spe-cies competition. Molecular evidence suggests that the predator-prey arms races shape the body plan and diversifi-cation of modern cephalopods during the Mesozoic Marine Revolution [40,41]: Mighty predators and niche compet-itors spurred slow ancient cephalopods to abandon their heavy shells and acquire competitive mobility [42,43]. As they dominate nearly the whole ocean and flow into nearshores, optically dynamic environments urged loliginids (estimated ~150 million years ago) to evolve tunable dermal iridescence [14,44]. As a behavioral strategy, the ven-tral reflector directs light into the surrounding seawater to mimic down-welling moonlight and starlight. It can puzzle predators and prevent attack from the blow [13,45,46]. Meanwhile, the sexual dimorphism of iridescence in squids, including features (location and tenability), is likely to serve as significant signals during mating or repro-duction [4].”
Comment: Line 116-122: is redundant and should be removed or condensed.
Reply: These sentences have been removed.
Comment: Line 144-147: the relevance of this section to the subsequent section and to iridophores is unclear.
Reply: We have rewrited the sentence “The evolvement of reflectins’ block structure and conserved reflectin motifs has been recently depicted, with evidence provided by symbiotic bacteria Vibrio fischeri [55].”.
Comment: The authors have several instances of single sentence paragraphs (see line 156-158), which make the text challenging to read. These “paragraphs” should be combined with the paragraphs before them to enhance the overall readability of the manuscript.
Reply: Single sentence has been removed or fused into other paragraphs.
Comment: Line 201-210 and 211-218 is redundant and unnecessary
Reply: 201-210 has been removed. 211-218 has been simplified.
Comment: Should combine lines 227 and 228.
Reply: Paragraphs have been combined.
Comment: Transitions required before line 284, line 291
Reply: We have changed the statement of 284 “More conceptually, Gorodetsky et al. abandoned the employment of reflectins and developed a TiO2/SiO2 Bragg stack–modified device with the low and high indices of refraction.” and 291 “Likewise beyond the color-change fields, the bulk protonic conductivity of reflectins and reflectin-derived polypep-tides is also discovered, which is a pioneered attempt at developing protein-based protonic transistors [34,35].”
Comment: Move 306-310 to conclusion
Reply: 306-310 has been moved.
Comment: Line 389-414 is redundant and should be removed from the conclusion.
Reply: We agree that the description is redundant. However, the deletion of these paragraphs will destroy the integrity of a conclusion. So we have simplified the sentences and combined the two paragraphs:” Reflectins have unusual amino-acid compositions and sequential properties, which endow themselves with properties including an extremely high reflective index (1.591±0.002), and responsiveness to various environmental stimuli [18]. Based on their unique material compositions and self-organization properties, reflectin-derived pep-tides, and purified reflectins are fabricated into an impressive array of thin films, fibers, diffraction gratings, and complex multi-layered optical systems [5,23,25,29,33,59]. More recently, mammalian cells are creatively used as platforms to explore the function potentials of reflectin proteins and investigate their working mechanism in the cellular environment [36,37].”
Comment: The authors introduce new concepts in line 433. This is unnecessary in the conclusion and should be discussed in the main body.
Reply: Relevant sentences have been moved to Part 5. Exploring reflectins’ functions and behaviors in cells.
Comment: Line 436-438 This statement does not conclude the text and is generally difficult to follow – what are polarized cells and what part of reflectin research discussed in this paper has any bearing on the field? Rather than focusing on applications from single studies, the authors should provide broader applications for the findings obtained from reflectin studies both in vivo and in vitro.
Reply: Lines 436-438 have been moved to Part 5. Exploring reflectins’ functions and behaviors in cells, with new statement” As a suggestion and a plan for what we are working on, cephalopod skin cells or polarized cells with highly orga-nized cytoskeleton can be employed as platforms to establish more delicate and effective reflectin-based photonic structures, since their sophisticated cytoskeleton system will provide reflectins with effective transportation and distribution networks.”

Round 3
Reviewer 1 Report
The authors have sufficiently addressed my comments. I would recommend extensive English corrections to enhance the readability of the manuscript, but otherwise would recommend publication of the manuscript.